# Mitochondrial Aldehyde Dehydrogenase 2 (ALDH2) Protects against Binge Alcohol-Mediated Gut and Brain Injury

**DOI:** 10.3390/cells13110927

**Published:** 2024-05-28

**Authors:** Bipul Ray, Wiramon Rungratanawanich, Karli R. LeFort, Saravana Babu Chidambaram, Byoung-Joon Song

**Affiliations:** 1Section of Molecular Pharmacology and Toxicology, National Institute on Alcohol Abuse and Alcoholism, Bethesda, MD 20892, USA; bipul.ray@nih.gov (B.R.); wiramon.rungratanawanich@nih.gov (W.R.); karli.lefort@nih.gov (K.R.L.); 2Department of Pharmacology, JSS College of Pharmacy, and Center for Experimental Pharmacology and Toxicology, JSS Academy of Higher Education & Research, Mysuru 570015, India; babupublications@gmail.com

**Keywords:** aldehyde dehydrogenase 2, mitochondrial ALDH2, alcohol, gut leakiness, brain damage, gut–brain axis, oxidative stress, post-translational modifications

## Abstract

Mitochondrial aldehyde dehydrogenase-2 (ALDH2) metabolizes acetaldehyde to acetate. People with ALDH2 deficiency and *Aldh2*-knockout (KO) mice are more susceptible to alcohol-induced tissue damage. However, the underlying mechanisms behind ALDH2-related gut-associated brain damage remain unclear. Age-matched young female *Aldh2*-KO and C57BL/6J wild-type (WT) mice were gavaged with binge alcohol (4 g/kg/dose, three doses) or dextrose (control) at 12 h intervals. Tissues and sera were collected 1 h after the last ethanol dose and evaluated by histological and biochemical analyses of the gut and hippocampus and their extracts. For the mechanistic study, mouse neuroblast Neuro2A cells were exposed to ethanol with or without an Aldh2 inhibitor (Daidzin). Binge alcohol decreased intestinal tight/adherens junction proteins but increased oxidative stress-mediated post-translational modifications (PTMs) and enterocyte apoptosis, leading to elevated gut leakiness and endotoxemia in *Aldh2*-KO mice compared to corresponding WT mice. Alcohol-exposed *Aldh2*-KO mice also showed higher levels of hippocampal brain injury, oxidative stress-related PTMs, and neuronal apoptosis than the WT mice. Additionally, alcohol exposure reduced Neuro2A cell viability with elevated oxidative stress-related PTMs and apoptosis, all of which were exacerbated by Aldh2 inhibition. Our results show for the first time that ALDH2 plays a protective role in binge alcohol-induced brain injury partly through the gut–brain axis, suggesting that ALDH2 is a potential target for attenuating alcohol-induced tissue injury.

## 1. Introduction

Alcohol (ethanol, EtOH) is mostly metabolized in the liver by cytosolic alcohol dehydrogenase (ADH) and mitochondrial aldehyde dehydrogenase-2 (ALDH2), both of which are regulated by various genetic and environmental factors in humans [1]. Under excessive amounts of alcohol intake, the ethanol-inducible cytochrome P450-2E1 (CYP2E1) localized in microsomes and mitochondria is known to metabolize EtOH to toxic acetaldehyde [2]. Mitochondrial ALDH2 is the key enzyme that converts acetaldehyde to acetate. A major genetic mutation in the *ALDH2* gene (i.e., the *ALDH2*2* variant allele) with decreased activity is found in almost one-third of East Asian people [3,4]. Individuals with the *ALDH2*2* single-nucleotide polymorphism (SNP) likely display a number of symptoms after drinking alcohol due to a greater buildup of acetaldehyde than those without this SNP. These uncomfortable symptoms include facial flushing, hyper-palpation, tachycardia, nausea, headaches, and sweating [5]. Consequently, many people with the *ALDH2*2* variant allele do not usually drink alcohol. Studies on populations with the *ALDH2*2* variant gene or *Aldh2*-knockout (KO) mice have shown greater sensitivity to alcohol-induced multiple organ damage than the control counterparts, resulting in conditions such as alcohol-related polyneuropathy [6], aging-related neuronal degeneration, and cognitive impairments [7]. In fact, deficiency in the ALDH2 enzyme increases the risk for late-onset Alzheimer’s disease [8,9,10,11] and cancer in multiple organs [12,13] in several populations. Thus, the *ALDH2* gene has been known to be a protective or alcohol-deterrent gene. This increased susceptibility to alcohol-mediated various tissue damage could be due to the accumulation of toxic acetaldehyde and possibly other reactive lipid aldehydes, including acrolein (ACR) [14,15].

It is well-established that binge-drinking alcohol causes gut leakiness (with physical disruption of the intestinal villi), leading to the translocation of pathogenic bacteria and endotoxins (lipopolysaccharide, LPS) released from Gram-negative bacteria to the circulation, leading to systemic inflammation and various tissue injuries [16,17,18]. Alcohol-induced gut and brain injury through the gut–brain axis with elevated gut dysbiosis and serum LPS in animal models and people with alcohol misuse has been reported by multiple laboratories [19,20,21,22,23]. It is also known that LPS alone can cause neuronal cell death in in vivo and in vitro models [24,25,26]. It is now widely reported that gut dysbiosis can affect central nervous system (CNS) function via the gut–brain axis [27,28]. LPS can trigger the immune cells in the gut, causing systemic inflammation with elevated TNF-α, which stimulates cell death signals and post-translational modifications (PTMs), including the nitration of intestinal tight junction (TJ) and adherens junction (AJ) proteins, resulting in their ubiquitin-dependent proteolysis and subsequent gut leakiness. Gut leakiness further allows the translocation of endotoxin (LPS) and other toxic molecules to the systemic circulation, interfering with immune function and causing damage to many tissues, including the brain, in a vicious cycle [27,29].

While ALDH2 is a key enzyme for acetaldehyde metabolism in the liver, ALDH2 is also present in the brain and other organs [1,30]. ALDH2 is expressed in the frontal and temporal cortices, primarily in glial cells and neuropils in the mid-brain, and in the hippocampus [31]. ALDH2 is also reported to be neuroprotective against various neurodegenerative diseases such as Alzheimer’s disease and Parkinson’s disease [7,32,33,34], possibly by removing highly reactive and cytotoxic lipid aldehydes such as 4-hydroxynonenal (4-HNE), malondialdehyde (MDA), and ACR. In this study, we aimed to investigate the protective role of ALDH2 in binge alcohol-induced gut and brain injury by comparing *Aldh2*-KO versus WT mice upon exposure to binge alcohol. We hypothesized that binge EtOH exposure causes greater brain damage in *Aldh2*-KO mice compared to their WT counterparts through elevated gut leakiness and endotoxemia. To our knowledge, there are no reports on the role of ALDH2 in gut-leakiness-associated brain injury through the gut–brain axis following binge alcohol exposure.

## 2. Materials and Methods

All animal experimental procedures (protocol no. LMBB-BS-1 approved on 27 February 2023) were carried out following the National Institutes of Health (NIH) guidelines for small animal experiments and approved by the NIAAA Institutional Animal Care and Use Committee. All mice were maintained under a controlled environment: temperature (22 ± 3 °C), humidity (45–55%), and tightly controlled lighting (12 h light/dark cycle) with free access to food and water provided ad libitum. Young female, age-matched inbred global *Aldh2*-KO mice with a C57BL/6J background and WT mice (n ≥ 9–12/group) [35] were orally (via gavage) exposed to three consecutive doses of binge alcohol (4 g/kg/dose) at 12 h intervals. In our previous report [17], we treated mice and rats with three consecutive doses of binge alcohol (5 g/kg/dose) at 12 h intervals to demonstrate the contributing role of CYP2E1 in alcohol-induced gut and liver injury. However, in the current study, we chose a slightly lower alcohol dosage (4 g/kg/dose at 12 h intervals) since we expected that *Aldh2*-KO mice would be more sensitive to alcohol-induced tissue injury than the corresponding WT mice. Control mice were orally administered dextrose to match the energy balance of EtOH administration. The intestinal enterocytes, blood, and brain tissue were collected at 1 h after the last dose of EtOH administration and stored at −80 °C until further characterization.

### 2.1. Histological H&E and TUNEL Staining

Sections of the intestine (ileum) from alcohol-exposed or control mice were fixed in 10% neutral-buffered formalin for histopathological analysis. Paraffin-embedded blocks of formalin-fixed individual intestine (ileum near the cecum) sections were cut at 4 μm and stained with hematoxylin and eosin (H&E) from American Histolabs Inc. (Gaithersburg, MD, USA). For TUNEL staining, formalin-fixed intestinal samples were processed, and 4 μm thick paraffin sections were used. The ApopTag Peroxidase in situ apoptosis detection kit (Millipore, Billerica, MA, USA) was employed to assess DNA strand breaks to evaluate the rates of apoptotic cell death, as described [35].

### 2.2. Measurements of Serum LPS, IL-6, TNF-α, and EtOH Contents and Caspase-3 Activity

Serum endotoxin (LPS) concentrations were measured by the quantitative endpoint Pierce™ Chromogenic Endotoxin Quant Kit (A39552). Serum IL-6 and TNF-α levels were evaluated by using the Abcam IL-6 ELISA Kit (ab285330) and TNF-alpha ELISA Kit (ab285327), respectively. Serum EtOH concentration was determined using the Abcam Ethanol Assay Kit (ab65343) by following the manufacturer’s protocol [35]. The Caspase-3 activity of gut enterocytes and brain hippocampus extracts was measured by following the manual of the Abcam Caspase-3 Assay kit (ab39401).

### 2.3. FITC-Dextran 4 kDa Analysis to Determine the Rate of Intestinal Permeability

To determine the rates of alcohol-induced intestinal permeability, FITC-D4 (25 mg/mL, Sigma-Aldrich, Saint Louis, MO, USA) was orally administered to all mice with the third (last) dose of EtOH or dextrose (control). The fluorescent emission of serum FIDC-D4 collected at 1 h post-ethanol gavage was measured with a microplate reader at excitation and emission spectra of 485 nm and 540 nm, respectively.

### 2.4. Fluoro-Jade-C Staining to Determine the Rate of Neurodegeneration

The frozen mid-brain tissues were sectioned to 10 µm using a Cryostat. The experiment was performed as per the instruction manual provided by the Histo-Chem Inc. Briefly, brain slides were incubated in 1% sodium hydroxide in 80% EtOH for 5 min. Slides were rinsed for 2 min in 70% EtOH, for 2–3 min in distilled water, and then incubated in 0.06% potassium permanganate solution for 10 min. Following a 1–2 min water rinse, the slides were then transferred for 10 min to a 0.0001% solution of Fluoro-Jade-C (F.-J.C.) dissolved in a 0.1% acetic acid vehicle. The slides were then rinsed 3 times with distilled water for 1 min each. Removing the excess water with tissue paper, the section slides were further kept on a slide warmer at 50 °C for at least 5 min. The air-dried slides were then cleared in xylene for at least 1 min and cover-slipped with Fluoromount-G™ Mounting Medium (Invitrogen™, Waltham, MA, USA). The fluorescence images were visualized by using a confocal microscope (Carl Zeiss LSM 700, Oberkochen, Germany).

### 2.5. Cell Culture, ALDH2 Activity Measurement, and Immunoblot Analysis

Neuro2A mouse neuroblast cells grown in a humidified incubator at 37 °C under 5% CO_2_ were pretreated with an ALDH2 inhibitor (Daidzin) at 20 or 40 µM for 12 h followed by exposure to 12.5 or 25 mM EtOH for an additional 12 h [35] before being harvested for further analyses. Protein concentrations were determined using the Bradford protein assay kit (Bio-Rad, Hercules, CA, USA). ALDH2 activity was evaluated using a Mitochondrial Aldehyde Dehydrogenase (ALDH2) Activity Assay Kit (Abcam, Cambridge, MA, USA) following the manufacturer’s protocol. For immunoblot analysis of Neuro2A cell extracts, equal amounts of protein from the indicated samples were resolved in 10~15% SDS-polyacrylamide gel electrophoresis, transferred to nitrocellulose membranes, and probed with the respective primary antibody against Aldh2 (1:1000 dilution; Abcam), Cleaved (active)-caspase-3 (1:1000 dilution; Cell Signaling, Danvers, MA, USA), 3-nitrotyrosine (3-NT) (1:1000 dilution; Santacruz), acrolein-protein adducts (1:1000 dilution; Abcam), acetylated-lysine (Ac-Lys) (1:1000 dilution; Cell Signaling), or Gapdh (1:5000 dilution; Cell Signaling), as described [35]. Appendix A containing the full-size original immunoblots for all IB images is included.

### 2.6. MTT Assay for Measurement of Neural Cell Viability

Neuro2A cells were treated with 12.5 or 25 mM EtOH in the presence or absence of an ALDH2 inhibitor (Daidzin) at 20 or 40 µM [35] for 12 h. After exposure to EtOH for 12 h, the cell medium was then replaced with a fresh culture medium for MTT treatment, and the cells were incubated at 37 °C for an additional 3 h. MTT solution (Abcam, Cambridge, MA, USA) was added, and the cells were further incubated at room temperature for 15 min before the absorbance at OD 590 nm was measured with a microplate reader, as previously reported [35].

### 2.7. Immunoblot Analysis of Gut Enterocytes and Hippocampal Extracts

The enterocytes of the small intestine and hippocampus from each mouse were homogenized with 1× RIPA buffer to prepare gut or hippocampal extracts. Protein concentrations of pooled extracts within the same groups (n ≥ 6~8/group) were determined using the Bradford protein assay kit, and equal amounts of protein from different group samples were separated by 10% SDS/PAGE and transferred to nitrocellulose membranes. These membranes were probed with the respective primary antibody against Claudin-1 (1:1000 dilution; Abcam), Occludin (1:1000 dilution; Abcam), β-catenin (1:1000 dilution; Abcam), E-cadherin (1:1000 dilution; Cell Signaling), Aldh2 (1:1000 dilution; Abcam), Cleaved (active)-caspase-3 (1:1000 dilution; Cell Signaling), Gapdh (1:5000 dilution; Cell Signaling), 3-nitrotyrosine (3-NT) (1:1000 dilution; Santacruz), Acrolein-protein adducts (1:1000 dilution; Abcam), Ubiquitin (1:1000 dilution; Cell Signaling), Ac-Lys (1:1000 dilution; Cell Signaling), or Ac-α-Tubulin (Lys40) (1:1000 dilution; Cell Signaling). Horseradish peroxidase (HRP)-conjugated anti-rabbit or anti-mouse IgG (1:5000 dilution; Cell Signaling) was used as the secondary antibody. Relative protein images were assessed by enhanced chemiluminescence (ECL) substrates, and their immunoreactive band intensities were quantified by densitometry using ImageJ software (version 1.8, National Institutes of Health, Bethesda, MD, USA), as described [35].

### 2.8. Statistical Analysis

Statistical significance was determined by two-way or one-way ANOVA and Tukey’s post-hoc multiple comparisons test to compare the means of multiple groups using Graph Pad Prism version 8.0 (Graph Pad Software, San Diego, CA, USA). Data are presented as means ± SD and considered statistically significant at *p* ≤ 0.05 [35].

## 3. Results

### 3.1. Binge Alcohol Exposure Caused Intestinal Disintegration and Increased Serum FITC-D4 and Endotoxin (LPS) Levels in Aldh2-KO Mice

Histological analysis of the gut ileum revealed greater intestinal damage with more short and blunted microvilli in alcohol-exposed *Aldh2*-KO mice compared to the corresponding WT counterparts (Figure 1A). However, similar levels of increased serum EtOH were observed in both *Aldh2*-KO and WT mice 1 h after alcohol exposure (Figure 1B). Alcohol significantly increased serum levels of FITC-D4 (Figure 1C), endotoxin LPS (Figure 1D), interleukin-6 (IL-6) (Figure 1E), and TNF-α (Figure 1F) in alcohol-exposed *Aldh2*-KO mice compared to corresponding WT mice. These results confirm the alcohol-induced loss of intestinal integrity in *Aldh2*-KO mice compared to WT mice, suggesting a protective role of ALDH2 against alcohol-mediated gut leakiness.

### 3.2. Binge Alcohol Exposure Elevated Intestinal Apoptosis, Oxidative Stress-Related PTMs, and Degradation of Gut TJ/AJ Proteins in Aldh2-KO Mice

After observing the evidence of enhanced leaky guts with intestinal damage and significantly elevated serum FITC-D4 and LPS levels in alcohol-exposed *Aldh2*-KO mice compared to WT mice, we performed an additional study to find out the underlying mechanism of alcohol-induced gut leakiness. Our results revealed that binge alcohol exposure significantly increased apoptosis, confirmed by TUNEL staining (Figure 2A), its quantification (Figure 2B), and Caspase-3 activity measurement (Figure 2C), a well-established marker of apoptosis, in the gut enterocytes of *Aldh2*-KO mice compared to the corresponding WT mice.

We hypothesized that increased oxidative stress and enterocyte apoptosis lead to various PTMs, followed by the proteasomal degradation of TJ/AJ proteins that maintain the integrity of the intestinal barrier. Extracts of isolated enterocytes were further analyzed to determine the levels of the indicated gut proteins (Figure 3A). Immunoblots showed that alcohol exposure significantly increased the levels of an apoptosis marker, cleaved (activated) caspase-3, and oxidative stress-related PTMs, including 3-NT, Ac-Lys, ubiquitin, and toxic acrolein-protein adducts, in *Aldh2*-KO mice compared to WT mice. However, the amounts of some proteins were significantly altered in WT mice after alcohol exposure, as indicated (Figure 3A). These findings are consistent with the results of biochemical and histological analyses for increased gut injury in the alcohol-exposed *Aldh2*-KO mice (Figure 1 and Figure 2).

The levels of gut TJ/AJ proteins, such as Claudin-1, E-Cadherin, β-Catenin, Occludin, and α-Tubulin (Figure 3B), were significantly decreased in alcohol-exposed *Aldh2*-KO mice. In contrast, much fewer changes were observed in the alcohol-treated WT mice. Previous reports showed that various PTMs, such as nitration [36] and acetylation [37], are associated with ubiquitin-dependent proteasomal degradation. A similar pattern of degradation was also observed in these proteins through increased nitration and acetylation in *Aldh2*-KO mice after binge alcohol administration compared to the WT counterparts [35]. In this study, we demonstrated the increased acetylation of α-Tubulin (Figure 3A) in alcohol-exposed *Aldh2*-KO mice compared to the WT counterparts. Elevated PTMs (nitration, acetylation, ubiquitination, etc.) of intestinal proteins were also observed in alcohol-administered *Aldh2*-KO mice. These data indicate that various PTMs promote the proteolytic degradation of cellular proteins [36,37], including gut TJ/AJ proteins [38].

### 3.3. Binge Alcohol Exposure Induced Neurodegeneration, Oxidative Stress, and Neuronal Apoptosis in Aldh2-KO Mice

After confirming the protective role of ALDH2 in alcohol-mediated gut injury, we further investigated the effect of binge alcohol on brain damage in *Aldh2*-KO mice compared to the WT counterparts. To investigate the brain damage, we prepared cryosections of the mid-brain for staining with Fluoro-Jade C (F.-J.C.), which is widely used to identify degenerative neuronal cells, including apoptotic and necrotic cells [39]. However, F.-J.C. may also detect non-neuronal cells, as reported [40]. In our confocal microscopy analysis, the DAPI-stained slides were used as positive controls to stain all neuronal cells. The hippocampus section was found to have more F.-J.C.-positive cells in binge alcohol-administered *Aldh2*-KO mice compared to the dextrose-treated WT counterparts, although fewer F.-J.C.-positive cells were observed in alcohol-exposed WT mice (Figure 4). These data suggest that binge alcohol exposure caused gut leakiness and brain damage in *Aldh2*-KO mice, at least partly through gut–brain interactions.

To further strengthen the evidence, hippocampus tissues from different mouse groups were homogenized to prepare the extracts for immunoblot analysis. Supporting the F.-J.C. confocal image data, oxidative stress-related acrolein-protein adducts, cleaved caspase-3, and activity of caspase-3, as markers of apoptosis (Figure 5), were found to be significantly higher in alcohol-exposed *Aldh2*-KO mice when compared to their WT counterparts. These results support the protective role of ALDH2 in binge alcohol-induced brain injury, as observed in *Aldh2*-KO mice.

### 3.4. ALDH2 Suppression Enhanced Oxidative Stress-Mediated PTMs and Apoptosis of Neuronal Cells

To further study the protective mechanisms of ALDH2 against alcohol-induced neuronal damage, Neuro2A neuroblast cells were pretreated with an ALDH2 inhibitor (Daidzin) at 20 or 40 µM for 12 h followed by exposure to 12.5 or 25 mM EtOH for an additional 12 h. Exposure to EtOH at a concentration of 25 mM alone decreased cell viability, and this was further reduced in the presence of Daidzin at 40 µM (Figure 6A). Therefore, pretreatment with Daidzin at 40 µM followed by exposure to EtOH at 25 mM was utilized for additional experiments. Figure 6B represents the immunoblot results of the indicated protein expression. The ALDH2 protein levels and activity (Figure 6C) were slightly decreased by EtOH exposure but significantly suppressed by Daidzin treatment. Consistent with the gut and brain of Aldh2-KO mice (Figure 4 and Figure 5), EtOH alone induced oxidative stress-related PTMs (including 3-NT, Ac-Lys, and Acrolein-adducts) and neuronal cell apoptosis, most of which were exacerbated by the suppressed ALDH2 after Daidzin treatment (Figure 6B,C). These findings support the protective role of ALDH2 in alcohol-mediated oxidative PTMs and neuronal cell damage, similar to the increased brain damage observed in alcohol-exposed Aldh2-KO mice.

## 4. Discussion

People with the *ALDH2*2* variant allele are known to be more susceptible to alcohol- or age-related neuronal damage [6,7,8,9,10,11] or cancer [12,13] compared to their unaffected counterparts. However, the underlying mechanisms remain unclear. Many epidemiological studies and experimental reports showed that excessive alcohol intake increases the risk of damage to various tissues such as the liver, gut, and kidneys [41,42,43], and could also cause an irreversible impact on the brain [44,45,46]. Many laboratories, including ours, reported that binge alcohol intake causes gut leakiness, which is likely to be associated with damage to tissues including the gut, liver, and brain through the gut–liver [16,17,18,47] or gut brain axes [27,29]. Binge alcohol was shown to induce gut leakiness by decreasing the levels of the intestinal TJ (e.g., occludin, claudin-1) and AJ (e.g., β-catenin, E-cadherin, α-tubulin) proteins via oxidative stress-related PTMs such as nitration, acetylation, and ubiquitination [17,48]. Gut leakiness allows pathogenic bacteria and their by-products, such as endotoxin (LPS), to enter the systemic circulation to alter innate immune function and promote end-organ damage, including damage to the brain [43,47].

ALDH2 plays a crucial role in metabolizing and clearing reactive aldehydes, including acetaldehyde, in the liver, gut, and other tissues [35,49,50]. For instance, the absence of ALDH2 through genetic mutation or its suppression in different conditions enhances oxidative stress-related damage to multiple organs, including the brain [46,47,48,49]. It is well-established that people with the *ALDH2*2* variant gene have virtually no ALDH2 activity and are very susceptible to tissue injury or carcinogenesis [12,13] after alcohol intake, due to the accumulation of acetaldehyde [1] and other toxic lipid aldehydes [51,52,53]. Approximately 35–45% of the total East Asian population has an *ALDH2*2* polymorphism, and likely accumulates acetaldehyde in the blood, liver, and brain upon alcohol intake [13,52,54]. Accumulation of toxic acetaldehyde and other potentially harmful lipid peroxides, including MDA, 4-HNE, and acrolein, increases susceptibility to alcohol-induced gut leakiness and tissue injury [51,52,53]. Recently, our laboratory has reported that a single-dose ethanol binge (as low as 3.5 or 4 g/kg) induces gut leakiness, endotoxemia (LPS), and acute liver injury in *Aldh2*-KO mice in comparison to the corresponding WT counterparts [35]. To our knowledge, the protective role of ALDH2 against binge alcohol-related brain damage through gut leakiness has not been studied so far. Gut leakiness promotes the transportation of gut bacteria and their products such as LPS to the systemic circulation to promote damage to tissues such as the liver, kidneys, and brain [26,55]. However, the exact mechanisms underlying damage to the gut–brain axis are not yet clear. In the current study, we hypothesized that three doses of binge alcohol at 12 h intervals are sufficient to cause brain damage in *Aldh2*-KO mice but not in the WT counterparts. The absence of mitochondrial ALDH2 reduces the capacity to detoxify elevated reactive aldehydes, including 4-HNE, MDA, and acrolein, since oxidative stress and increased levels of 4-HNE and acrolein can cause tissue damage, including in the gut and brain [49,50,51,56,57].

Here, we have provided evidence of alcohol-induced sensitivity to tissue damage via ALDH2 deficiency, where binge alcohol exposure showed higher gut leakiness and brain injury in *Aldh2*-KO mice compared to the corresponding WT counterparts. After exposure to three consecutive doses of EtOH, serum FITC-D4 and LPS levels were significantly increased in *Aldh2*-KO mice compared to their WT counterparts, despite similar levels of serum EtOH concentrations in both mouse strains. We and other laboratories have shown that *Aldh2*-KO mice show high levels of acetaldehyde after a single ethanol dose in the blood, liver, and other tissues, including the brain [35,54]. Elevated acetaldehyde, the toxic ethanol metabolite, was shown to alter the distribution of intestinal TJ/AJ proteins, leading to intestinal epithelial barrier dysfunction in mice [55]. In addition, *Aldh2*-KO mice showed an accumulation of other lipid peroxides, including acrolein and MDA, after consuming a Western-style high-fat diet [58]. Accumulation of these endogenous cytotoxic aldehydes and inhibition of the ALDH2 enzyme by these reactive lipid aldehydes increase vulnerability to aldehyde-induced tissue damage [59], especially in one of its primary sites of metabolism in the liver [60].

The present finding shows that ALDH2 deficiency increased LPS and various PTMs such as nitration, acetylation, acrolein-adducts, and ubiquitination of gut TJ/AJ proteins, leading to tissue injury in alcohol-exposed *Aldh2*-KO mice, while few changes were observed in their WT counterparts. These current results are consistent with our previous reports [17,35,38], where intestinal TJ/AJ proteins were modified by various PTMs and decreased through ubiquitin-dependent proteolysis. In addition, alcohol consumption is frequently associated with both gut dysbiosis and bacterial overgrowth [61,62], which were shown to increase local inflammation and intestinal permeability with the release of endotoxins (LPS) to the systemic circulation and tissue injury [27,28]. Based on these results, it is likely that alcohol-mediated gut dysbiosis as shown in *Aldh2*-KO mice [35] may also contribute to increased intestinal barrier dysfunction and endotoxemia with elevated LPS, IL-6, and TNF-α, all of which can cause neuroinflammation [63] and brain damage in *Aldh2*-KO mice, at least partly through the gut–brain axis.

Importantly, mitochondrial ALDH2 is widely expressed in different brain regions, including the hippocampus [31,64,65]. Many potentially toxic agents, including alcohol [48,66,67,68], carbon tetrachloride [69], acetaminophen [70], and 3,4-methylenedioxymethamphetamine (Ecstasy) [71], can inhibit ALDH2 activity through oxidative PTMs of the Cysteine in its active site and other amino acid residues. Suppressed ALDH2 activity after alcohol exposure [48,66,67,68], or the presence of the *ALDH2*2* variant allele as observed in many East Asian people, likely increases susceptibility to organ damage [8,9,10,11,12]. In this study, we have reported increased neuronal damage in the hippocampus region in binge alcohol-exposed *Aldh2*-KO mice compared to their WT counterparts. These results are consistent with the increased sensitivity to alcohol-related neuropathy [6] and age-dependent neurodegeneration with cognitive impairments in *Aldh2*-KO mice compared to their WT counterparts [7]. Our current results indicate that intestinal hyper-permeability and increased endotoxin (LPS) levels, which activates neuroinflammation with elevated pro-inflammatory cytokines [63], likely play key roles in increasing systemic inflammation and neuronal damage in binge alcohol-exposed *Aldh2*-KO mice. Consistently, our in vitro data with Neuro2A cells also confirmed the increased levels of cleaved caspase-3 and oxidative stress-related PTMs after EtOH exposure. These changes were aggravated by suppressed ALDH2 activity (through Daidzin treatment), suggesting the significant role of ALDH2 in alcohol-induced neuronal cell damage.

## 5. Conclusions

These results show for the first time that ALDH2 plays a protective role in binge alcohol-induced gut and brain injury partly through the gut–brain axis and suggest that ALDH2 could be a potential target for preventing or treating alcohol-induced gut and brain damage.

## Figures and Tables

**Figure 1 cells-13-00927-f001:**
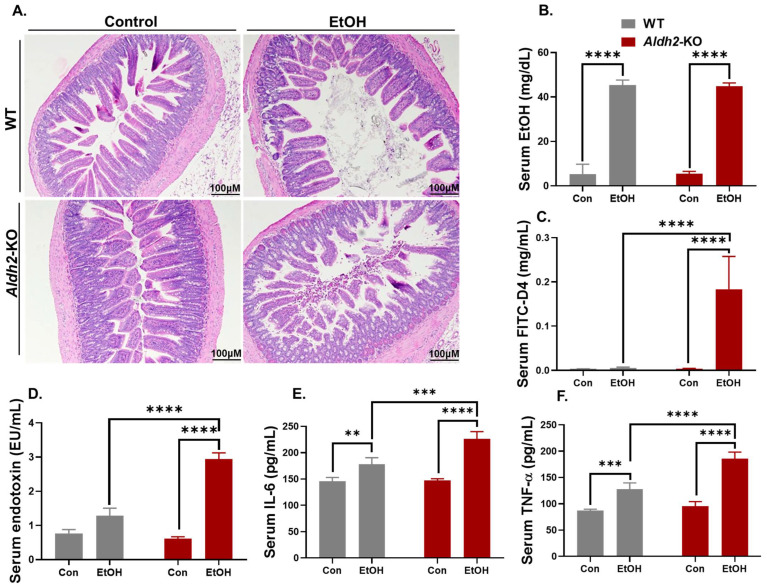
Binge alcohol exposure caused more intestinal disintegration and increased serum FITC-D4 and endotoxin (LPS) levels in *Aldh2*-KO mice compared to WT mice. (**A**) Representative H&E-stained histology images showed more ruptured intestinal villi structures in alcohol-exposed *Aldh2*-KO mice than corresponding WT mice, (**B**) despite the similar levels of serum EtOH concentrations between both mouse strains 1 h after alcohol exposure. (**C**–**F**) Binge alcohol exposure significantly increased the levels of (**C**) serum FITC-D4, (**D**) endotoxin (LPS), (**E**) IL-6, and (**F**) TNF-α in *Aldh2*-KO mice compared to the WT counterparts (samples from n = 3~5/group). Data were analyzed by two-way ANOVA, where ** *p* < 0.01, *** *p* < 0.001, and **** *p* < 0.0001.

**Figure 2 cells-13-00927-f002:**
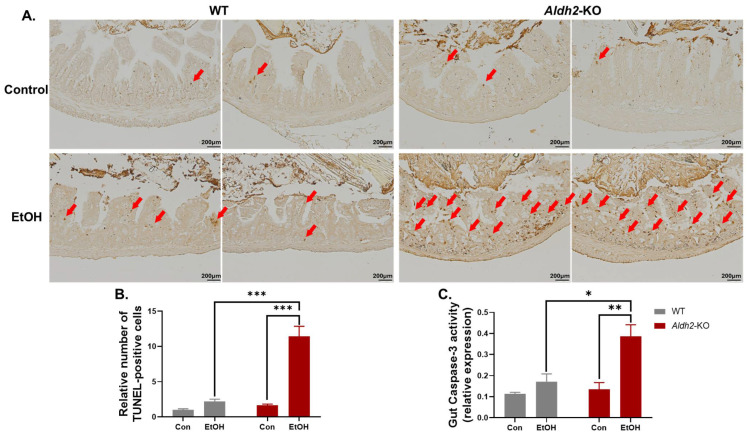
Binge alcohol significantly elevated intestinal apoptosis in *Aldh2*-KO mice compared to WT mice. (**A**) Representative images of TUNEL staining where the red arrows indicate TUNEL-positive cells, (**B**) quantification of TUNEL-positive cells, and (**C**) Caspase-3 activity assay results of the enterocyte extracts from the designated groups, revealing significantly higher levels of enterocyte apoptosis in alcohol-exposed *Aldh2*-KO mice than in the WT counterparts (n = 3~5/group). Data were analyzed by two-way ANOVA, where * *p* < 0.05, ** *p* < 0.01, and *** *p* < 0.001.

**Figure 3 cells-13-00927-f003:**
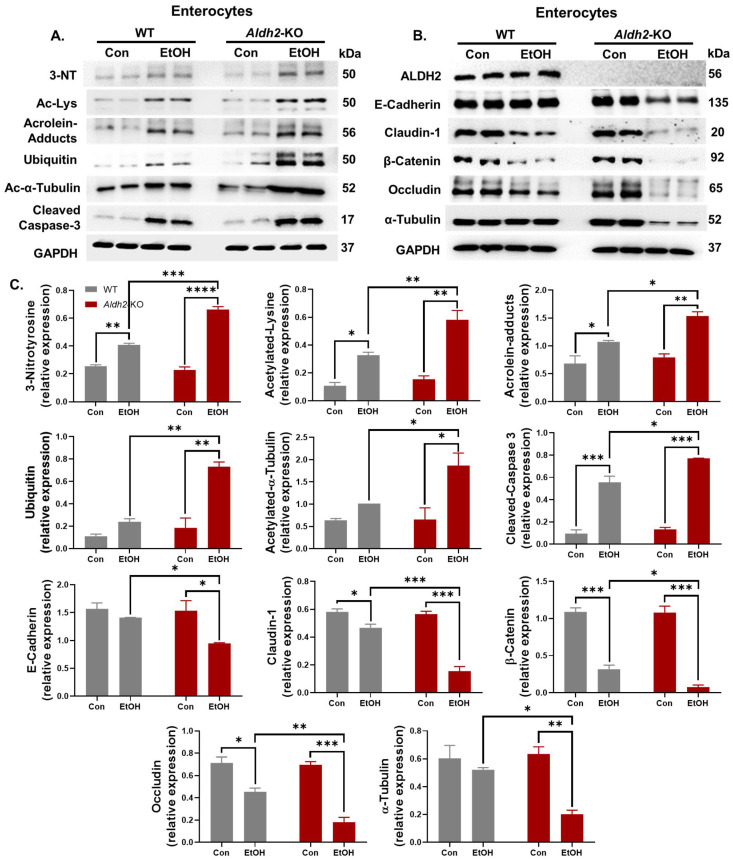
Binge alcohol exposure elevated intestinal apoptosis, oxidative stress-related PTMs, and degradation of gut TJ/AJ proteins in *Aldh2*-KO mice. (**A**) Representative immunoblot images of oxidative stress, PTMs, and apoptosis marker proteins in the gut extracts, as indicated. (**B**) Representative immunoblot images of intestinal TJ/AJ proteins, E-Cadherin, Claudin-1, β-Catenin, Occludin, and α-Tubulin, along with Aldh2 and Gapdh, for the indicated mouse groups (extracts from n = 3~5/group). (**C**) Densitometric quantitation of each protein level compared to Gapdh loading controls. The data were analyzed by two-way ANOVA, where * *p* < 0.05, ** *p* < 0.01, *** *p* < 0.001, and **** *p* < 0.0001.

**Figure 4 cells-13-00927-f004:**
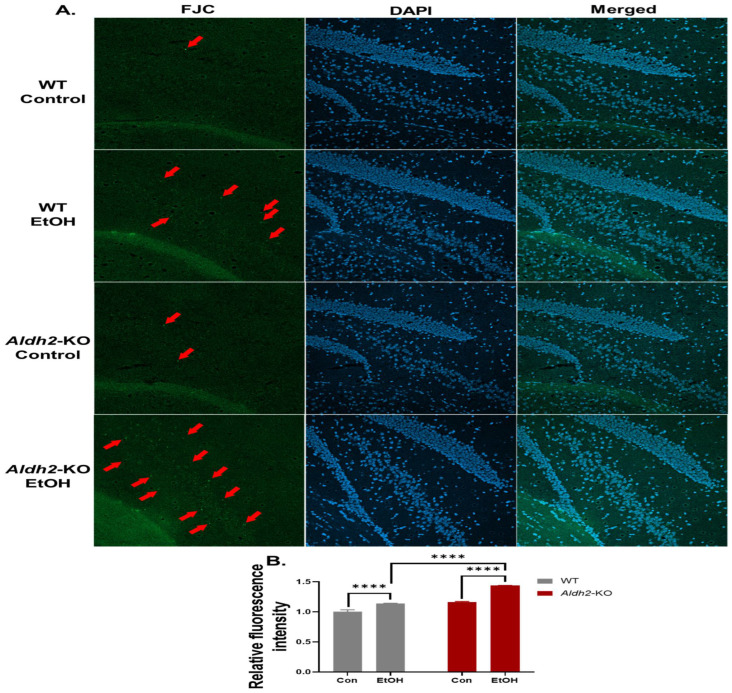
Binge alcohol increased brain damage in *Aldh2*-KO mice compared to WT mice. (**A**) The damaged neuronal cells detected by F.-J.C. staining were shown in bright green, as some were marked with red arrows in the hippocampus dentate gyrus region of mouse brains (n = 3/group, two separate preparations). DAPI was used to stain all neuronal cells. (**B**) Fluorescence intensity quantification. Data were analyzed by two-way ANOVA, where **** *p* < 0.0001.

**Figure 5 cells-13-00927-f005:**
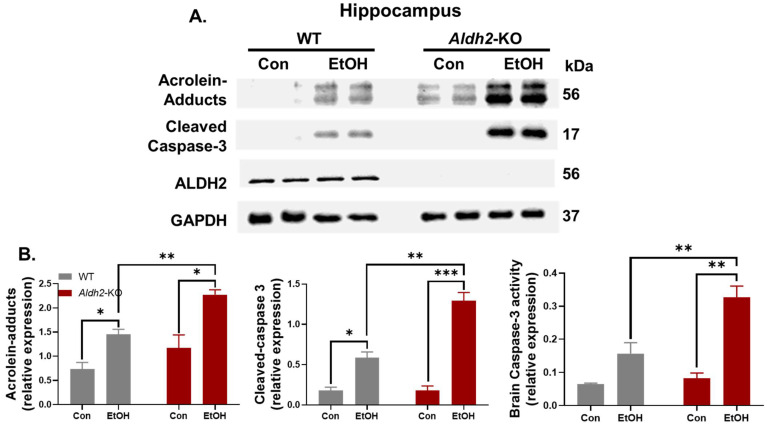
Binge alcohol exposure significantly increased the levels of acrolein-protein adducts and cleaved caspase-3, along with Caspase-3 activity, in the hippocampus extracts of alcohol-exposed *Aldh2*-KO mice compared to the WT counterparts (extracts from n = 3~5/group). (**A**) Representative immunoblot images and (**B**) densitometric quantitation of the indicated hippocampal protein, compared to Gapdh loading controls, and caspase-3 activity. These results were analyzed by two-way ANOVA, where * *p* < 0.05, ** *p* < 0.01, and *** *p* < 0.001.

**Figure 6 cells-13-00927-f006:**
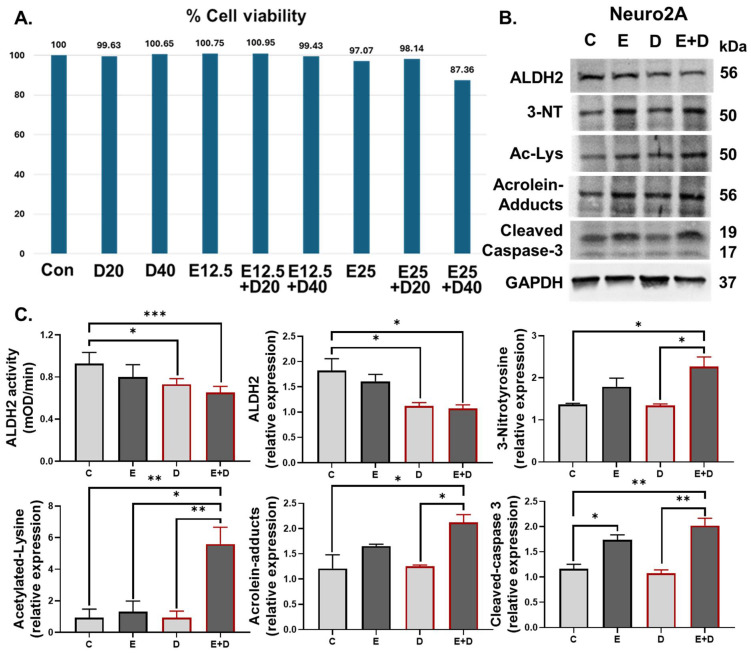
ALDH2 suppression enhanced oxidative PTMs and apoptosis of Neuro2A cells. (**A**) The rates of cell viability in Neuro2A cells exposed to an ALDH2 inhibitor (Daidzin) at 20 or 40 µM and 12.5 or 25 mM EtOH. (**B**) Representatives immunoblot images of ALDH2 and oxidative PTMs. (**C**) ALDH2 activity and densitometric quantitation of the indicated protein relative to Gapdh loading controls (extracts from n = 3–5/group, repeated twice). Data were analyzed by one-way ANOVA, where * *p* < 0.05, ** *p* < 0.01, and *** *p* < 0.001.

## Data Availability

Raw data are available in the lab as per NIH guidelines and will be made available upon request.

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
