# Peer review of "Mitochondrial Aldehyde Dehydrogenase 2 (ALDH2) Protects against Binge Alcohol-Mediated Gut and Brain Injury"

_cells, 2024, doi:10.3390/cells13110927_

Round 1

Reviewer 1 Report

Comments and Suggestions for Authors

In this study, the role of ALDH2 on EtOH-induced neuronal damage is analysed using ALDH2-KO mice. It is postulated that ALDH2 is significantly involved in the regulation of EtOH-induced effects by attenuating the gut-induced inflammatory response, which can ultimately lead to neuronal damage via the gut-brain axis. Intestinal changes in the stomach of mice after EtOH application were analysed. It was found that both intestinal permeability and serum endotoxin levels were increased in the ALDH2-KO mice. Apoptosis markers were also increased in the stomach extracts and intestinal sections, and proteins responsible for the permeability of the intestinal barrier were decreased. On the other hand, neuronal cell death was detected by Fluoro-Jade-C staining, as well as protein expressions associated with apoptosis. EtOH induced a similar pattern of apoptotic markers in the hippocampal tissue as well as in the Neuro2A cells both in vivo and in vitro.

The manuscript is well written, clearly presented and above all, shows very clear results. Beyond that, it is an appropriate work and underlines the importance of ALDH2 integrity, whose impaired function affects a significant number of people worldwide.

The most serious limitation of the presented study relies in the postulated relations between the gut-brain axis, the neuronal damage and the involvement of ALDH2 in this signalling axis.

In the Introduction, previous findings and the significance of the gut-brain-axis for alcohol-mediated brain damage are not sufficiently presented. Please state more precisely the importance of the gut-brain-axis for alcohol-mediated brain-damage.

As this relationship is not adequately defined, the results section does not allow for clear conclusions to be drawn about whether the observed effects of ALDH2 are indeed attributed to its influence on the gut-brain axis, especially considering that ALDH2 was ubiquitously knocked out in the animals.

This is further supported by the findings in the Neuro2A cells, which showed a similar pattern of damage to brain tissue from ALDH2-KO mice, independent of the gut-brain axis, by EtOH exposure only.

Have the parameters PTMs and apoptosis also been investigated in other tissue types from the mice, such as the liver, immune cells or other regions of the brain? This would help determine whether the effects attributed to ALDH2 are truly linked to the gut-brain axis or if they are merely systemic effects resulting from alcohol exposure.

In this context, the authors note that alcohol exposure impacts the intestine, leading to the translocation of LPS and other pro-inflammatory mediators, ultimately resulting in subsequent brain damage. To underscore the connection of the gut-brain axis, these inflammatory parameters, particularly TNF-α, should be analysed in the investigated brain regions.

Line 22-23: It is mentioned that tissues and sera were collected and evaluated by histological and biochemical analyses. Please add the information of tissue origin and details about the performed histological and biochemical analysis, it should be clearly stated what kind of experiments were performed.  

Line 23 and 28: It should be underlined from which tissue origin these cells are derived and which parts of the brain underwent apoptosis.

Line 24 and 30: In order to maintain a homogeneous format in the abstract, the subsections “Results” and “-Conclusion” should be avoided.

Line 218-219; 257-258; 312-314; 319-321: The sentence structure should be improved.

Line 232: Insert the figure number.

In order to better visualise the histological results, it is recommended to quantify Fig. 2A and Fig. 4.

Author Response

Please see the attached Word file for the point-by-point responses to the comments by the Editor and reviewers. Thanks.

Reviewer 2 Report

Comments and Suggestions for Authors

In the work entitled "Mitochondrial Aldehyde Dehydrogenase 2 (ALDH2) Protects Against Binge Alcohol-Mediated Brain Injury through the Gut-Brain Axis," Ray and colleagues investigate the underlying mechanisms for ALDH2-related gut-associated brain damage. By leveraging an Aldh2-knockout mouse model, the authors assess how binge alcohol exposure caused intestinal disintegration and increased serum levels of FITC-dextran 4 and endotoxin (LPS), induced neurodegeneration, oxidative stress, and neuronal apoptosis. Consistently, their in vitro data with Neuro2A cells also confirmed increased levels of cleaved caspase-3 and oxidative stress-related post-translational modifications (PTMs) after ethanol exposure. These changes were exacerbated when ALDH2 activity was suppressed (by daidzin treatment), suggesting the significant role of ALDH2 in alcohol-induced neuronal cell damage.

While this work is relevant, as the interaction between the gut and brain has gained interest in many disorders, the soundness of this work could be improved. Although the experimental design was well-performed, crucial information on the number of replicates (animals and experiments) is missing from the figures (Figures 1 through 6).

Author Response

(The authors gave the same response as above.)

Reviewer 3 Report

Comments and Suggestions for Authors

This is an intriguing study looking at the protective role of ALDH2 after binge alcohol exposure. The authors investigate effects on both the gut and the brain, noting that this axis is a critical determinant of outcomes following toxic alcohol exposure. Overall, the experimental design is sound, the results are well-explained, and the outcomes are logically considered within the context of the alcohol field. My major concern is with the lack of justification for the binge alcohol paradigm used, as well as some more minor issues regarding the reporting of blood alcohol levels and the timing of drug exposures in the Neuro2A in vitro model. Once these concerns are addressed, this manuscript could be a valuable addition to the field.

Specific concerns:

1) The binge alcohol paradigm consisted of three 4 g/kg doses, spaced 12 hours apart. The tissue analysis was then done 1 hour after the last of the three doses. The authors gave no rationale for this paradigm, nor were any previous studies cited that used the same approach. What is the translational relevance of this dosing strategy?

2) There is a discrepancy in the description of the Neuro2A dosing protocol. The Methods section implies that the ethanol and Daidzin treatments were performed simultaneously, but the Results section makes it sound like the treatments were done in sequence. Please clarify.

3) Error bars are said to be SD, but given the small variance of many of the graphs, I questioned whether they may be SEM instead. Are they SD or SEM?

4) Is there a way to quantify the "intestinal disintegration" shown in Figure 1A?

5) Figure 1B shows serum ethanol levels in nmols/uL, which is a fairly nonstandard reporting measure that has diminished translational value. An alternative unit such as mg/dL would be preferred.

6) In lines 214-215, the authors state that there were no significant changes seen in the WT mice, but the actual graphs state otherwise.

7) Fluoro-Jade C staining was performed to mark possible neurodegeneration, but this method is not neuron-specific (https://www.ncbi.nlm.nih.gov/pmc/articles/PMC8427928/). 

8) In Figure 6, for the majority of these measurements, the E+D values were not significantly different than E alone; this raises concerns about the authors' claims regarding ALDH2 inhibition exacerbating stress.

9) The Western blot results indicate that multiple blots were performed in order to visualize all of the antibodies that were probed, but only one blot was shown for GAPDH (which was used to normalize protein loading for each blot) for both the tissue and Neuro2A experiments.

Author Response

(The authors gave the same response as above.)

Round 2

Reviewer 2 Report

Comments and Suggestions for Authors

The authors addressed all the raised concerns.

Reviewer 3 Report

Comments and Suggestions for Authors

The authors have adequately responded to my previous criticisms and suggestions. I have no further issues.